# Overview of Inositol and Inositol Phosphates on Chemoprevention of Colitis-Induced Carcinogenesis

**DOI:** 10.3390/molecules26010031

**Published:** 2020-12-23

**Authors:** Samuel E. Weinberg, Le Yu Sun, Allison L. Yang, Jie Liao, Guang Yu Yang

**Affiliations:** 1Department of Pathology, Northwestern University Feinberg School of Medicine, 303 East Chicago Avenue, Chicago, IL 60611, USA; Samuel-Weinberg@northwestern.edu (S.E.W.); leyu.sun@northwestern.edu (L.Y.S.); jie-liao@northwestern.edu (J.L.); 2Division of Gastroenterology and Hepatology, Weill Cornell Medicine, 1293 York Avenue, New York, NY 10065, USA; aly4002@med.cornell.edu

**Keywords:** inositol, inositol phosphates, colitis, carcinogenesis, chemoprevention

## Abstract

Chronic inflammation is one of the most common and well-recognized risk factors for human cancer, including colon cancer. Inflammatory bowel disease (IBD) is defined as a longstanding idiopathic chronic active inflammatory process in the colon, including ulcerative colitis and Crohn’s disease. Importantly, patients with IBD have a significantly increased risk for the development of colorectal carcinoma. Dietary inositol and its phosphates, as well as phospholipid derivatives, are well known to benefit human health in diverse pathologies including cancer prevention. Inositol phosphates including InsP_3_, InsP_6_, and other pyrophosphates, play important roles in cellular metabolic and signal transduction pathways involved in the control of cell proliferation, differentiation, RNA export, DNA repair, energy transduction, ATP regeneration, and numerous others. In the review, we highlight the biologic function and health effects of inositol and its phosphates including the nature and sources of these molecules, potential nutritional deficiencies, their biologic metabolism and function, and finally, their role in the prevention of colitis-induced carcinogenesis.

## 1. Introduction

There is considerable evidence identifying a beneficial role of dietary inositol and its phosphates and phospholipid derivatives in human health [1,2,3]. In general, phospholipid derivatives are the most common form of the inositol compounds in the diet, and the amount of *myo*-inositol per 2500 kcal in the diet is approximately 900 mg [4]. Almost all of the ingested *myo*-inositol and its phosphates (99.8%) are absorbed from the human gastrointestinal tract [1,5]. There are at least nine isomers of inositol compounds, and the most common form is *myo*-inositol. *Myo*-inositol is a vitamin-like substance and can be produced de novo in the human body, particularly in the kidney [6,7]. In humans, there is approximately 0.03 mM concentration of *myo*-inositol in normal circulating fasting plasma and is thought to be quickly metabolized with a half-life of 22 min [1]. Compared to *myo*-inositol, approximately 79 ± 10% of its phosphate form, called inositol hexaphosphate (InsP_6_), is also rapidly absorbed and distributed widely throughout the body [5]. In this review, we highlight biologic functions and health effects of inositol and its phosphates from the nature and source, biologic metabolism and function, potential nutritional deficiency of inositol and its phosphates to the prevention of colitis-induced carcinogenesis.

## 2. Nature and Sources of Inositol Compounds 

Inositol has a similar chemical structure to glucose and is categorized as a carbohydrate. At least nine stereoisomers of inositol are identified and the difference of these inositol stereoisomers is the position of their -OH on the hexose ring. The nine inositol stereoisomers are *myo*-inositol, d-*chiro*-inositol, *scyllo*-inositol, *muco*-inositol, *neo*-inositol, l-*chiro*-inositol *cis*-inositol, *epi*-inositol, and *allo*-inositol. Among these isomers, *myo*-inositol is the most common in nature, and although d-*chiro*-inositol is naturally occurring, it occurs in minimal quantities in nature. *myo*-inositol exists either as a salt form such as inositol phosphates (InsPs), or a lipid form such as phosphatidylinositol (PI) and phosphatidylinositol phosphate (PIP) lipids, and is the most abundant inositol in eukaryotic cells [4,8].

The salt forms of inositol, known as inositol phosphates, are important biologic molecule/s. Mono- to polyphosphated inositols, including InsP_1_, InsP_2_, InsP_3_, InsP_4_, InsP_5_, InsP_6_, and inositol pyrophosphates, have diverse biologic functions and are known to act as signaling molecule/s in a variety of cellular processes including cellular proliferation, differentiation, migration, endocytosis, and apoptosis [2,3]. 

Mono- to polyphosphated inositols (InsP_1–8_) exist widely in nature. Although the specific nutritional value and detailed functions of these molecules are not fully understood, they comprise an important component of the human diet [4]. Approximately 900 mg of inositol-based compounds are present in the 2500 kcal North American diet. Lipid- and phosphate-forms (InsP_6_) of inositol are predominant [4,9]. Importantly, 99.8% (almost all) of ingested *myo*-inositol is absorbed from the human gastrointestinal tract with circulating fasting plasma *myo*-inositol concentration found to be approximately 0.03 mM. Importantly, *myo*-inositol is rapidly metabolized as evidenced by its half-life of 22 min [1].

A major lipid-form of inositol is inositol phosphatides, which is primarily acquired from the ingestion of lecithin. Lecithin is a component of dietary fat, commonly found in many foods, including soybeans and egg yolks [10]. Lecithin contains 20–21% inositol phosphatides that provides an important source of inositol. In general, lecithin is well absorbed and has relatively high bioavailability.

InsP_6_ is the most abundant form of inositol polyphosphates and exists ubiquitously in both mammalian cells and in plants with a concentration ranging from 10 to 100 µM [3,11]. A high-fiber diet is a major source of InsP_6_ [12]. Importantly, calcium magnesium InsP_6_ or “phytin,” the most common salt form of InsP_6_, is the major storage form of phosphorus and inositol in many plant tissues, particularly, in seeds such as nuts and grains, and especially in the bran, which provides major nutritional sources of inositol polyphosphates [13].

## 3. Biosynthesis and Metabolism of Inositol and Its Phosphates 

Inositol and inositol polyphosphates function as primarily as signaling molecules and secondary messengers [3,11]. Either phosphatidylinositol (an important lipid) or water-soluble inositol phosphates (a salt form) are involved in a number of biological processes and participates in essential metabolic processes in all plants and animals [2,8,13,14,15,16].

De novo biosynthesis of *myo*-inositol is a required cellular process. However, a significant quantity of whole-body inositol is acquired from the diet and transported into cells using a sodium-dependent process. Biosynthesis of *myo*-inositol in mammalian cells only occurs in the kidney in humans with a typical individual producing a few grams per day [17]. De novo *myo*-inositol synthesis occurs in two stages. Initially, glucose-6-phosphate (G6P) is first converted to *myo*-inositol-1-phosphate by inositol 1-phosphate synthase. Next, *myo*-inositol 1-phosphate is dephosphorlated to free *myo*-inositol by inositol monophosphatase [6,7,18]. This process requires an oxidized nicotinamide adenine dinucleotide (NAD) and is stimulated by NH_4_CL and MgCl_2_. The phosphorylation of inositol at different positions is performed by different kinases, while phosphates can be removed by a broad range of phosphatases. Thus, the balance of specific kinases and phosphates regulates the phosphorylation status of inositol, which has important ramifications for numerous biological process. These processes produce several significant signaling molecules such as *myo*-inositol (1,4,5)-trisphosphate (InsP_3_ or simply IP3) [3,11,14,19,20,21]. Free soluble InsP_3_ is produced through the hydrolysis of the phosphodiester of PI(4,5)P2 by phospholipase C (PLC) [22,23,24]. InsP_3_ is a crucial signaling molecule and second messenger that plays a central role in intracellular calcium release and regulation [25,26]. There are at least 63 possible isomers of phosphorylated inositol. Most of these isomers have been identified as metabolites in various biological systems, but the definitive functions of most of these molecules are not fully understood [27]. *Myo*-inositol (1,2,3,4,5,6)-hexakisphosphate (InsP_6_, also known as “phytate”) and *myo*-inositol(1,3,4,5,6)-pentakisphosphate (InsP_5_) are abundant in both mammalian and plant cells with concentrations ranging from 10 to 100 uM [28]. Various biological activities of InsP_6_ have been identified including roles in neurotransmission, immune responses, regulation of protein kinases and phosphatases, and activation of calcium channels [3,29]. InsP_6_ is synthesized from Ins(1,4,5)P3 in eukaryotic organisms through a sequential route involving several kinases and a phosphatase. The initial step is catalyzed by Ins(1,4,5)P3 3-kinase (InsP3K), followed by modification of Ins(1,3,4,6)P4 by inositol phosphate “multi-kinase” (IPMK) to Ins(1,3,4,5,6)P5 and Ins(1,3,4,5,6)P5 to Ins (1,2,3,4,5,6)P6 [3,29].

Inositol pyrophosphates (IPPs: InsP_7_ and InsP_8_), the complex form of inositol polyphosphates, have been identified recently in various biological systems including the yeast, *Saccharomyces cerevisiae*; amoeba, *Dictyostelium discoideum*; and mammalian cells. Chemically, IPPs are diphosphoinositol polyphosphates (PP-InsPs), and InsP_5_ and InsP_6_ serve as the precursors of these IPPs molecules and contain one or more pyrophosphate bonds [20]. ‘High energy’ is a key feature of PP-InsPs/IPPs, which are a critical energy source of cells. Energy metabolism such as ATP production is closely regulated by these pyrophosphate molecules and these inositol pyrophosphastes may function as master regulators of cellular energy metabolism [14]. The common pathway for inositol pyrophosphate synthesis is through phosphorylation of InsP_6_ by InsP6-kinase and PP-InsP5-kinases to produce diphoshoinositolpentakisphosphate (InsP_7_ or PP-InsP5) and bisdiphoshoinositoltetrakisphosphate (InsP_8_ or (PP)2-InsP4). This synthesis pathway is highly conserved process throughout evolution, and there are two distinct classes of enzymes, InsP6-kinase and PP-InsP5-kinases [27]. 

In addition to regulating energy metabolism, IPPs are involved in other diverse biologic process including apoptosis, autophagy, chemotaxis, embryonic development, telomere maintenance, etc. [14]. Interestingly, intracellular levels of InsP_7_ and InsP_6_ appears associated with oxidative stress based on the following evidence: (1) a reduction in InsP_7_ levels is observed in the yeast treated with hydrogen peroxide and (2) intracellular levels of InsP_7_ and InsP_6_ greatly differ in hepatocytes from the aged mice (>10-month-old) compared with young mice (<2-month-old) [14,30]. Importantly, alterations in IPPs have also been linked to immune function, insulin homeostasis, obesity etc. [14].

## 4. Potential Nutritional Deficiency of Inositol and Inositol Phosphates

Whether or not a nutritional deficiency in inositol and its phosphates occurs in humans is a critical question. The Western-like diet is rich in fat, sugars; yet is significantly deprived of fibers, which contain high amounts of inositol compounds. Thus, dietary deficiency in inositol and its phosphates likely can occur in humans [31]. The well-established epidemiological and clinical evidence demonstrates the protective role of a diet with a high content of fibers against the development of chronic diseases—including metabolic diseases (obesity, diabetes, etc.), cardiovascular diseases, and cancers [32,33,34]. Interestingly, early evidence suggests that phytic acid or InsP_6_ found in high-fiber food is a protective factor against the development of colorectal cancer [12]. Despite these findings, the importance of dietary inositol and its phosphates remains largely unexplored. The few available studies exhibit significant differences based on the country where the study was performed. In general, people in Western countries have the daily intake of inositol of < 500–700 mg/day, and higher consumption is seen in Africa and Asia. The average daily intake of InsP_6_ ranges from 170 to 390 mg/day for both the US and Canadian children [35,36], while the average intake in adults is significantly higher (averaging 538 mg/day) and shows a difference between males and females (608 mg vs. 512 mg/day, respectively) [37]. In comparison, adult Asian immigrants in Canada consume mostly a vegetarian, high-fiber diet with average daily InsP_6_ intake of 1487 ± 791 mg/day [38]. Overall, the evidence suggests that a deficiency in inositol and its phosphates may be associated with the increased risk of both metabolic and cancerous diseases [31].

Mechanically, either reduced dietary intake or metabolic impairments can affect the bioavailability of inositol in living organisms. Pharmacodynamic and pharmacokinetic studies of inositol(s) in humans demonstrate that high blood glucose levels increase inositol degradation and excretion and inhibits both *myo*-inositol biosynthesis and absorption [1,39]. These studies indicate that inositol levels are closely tied to the levels of other metabolites; thus, the nutritional importance and mechanism of *myo*-inositol and its phosphates in human health needs to be investigate further.

## 5. Chemopreventive Effects of Inositol and InsP_6_ on Cancer—Colitis-Induced Carcinogenesis as a Model

*Myo*-inositol and InsP_6_ have shown efficient anticancer action against various types of malignancies, including colon, mammary, lung, liver, etc. [2]. Inositol and InsP_6_ are common constituents of cereals and legumes. Importantly, a high-bran diet or equivalent doses of InsP_6_ alone both resulted in a significant reduction in the incidence of experimental colon and breast cancer [2,13,40]. Further, supplementation of pure inositol and InsP_6_ can enhanced a chemopreventive effect in preclinical experimental models [40,41,42,43,44,45]. Chronic inflammatory process in the colon, particularly, the typical inflammatory bowel diseases ((IBD), including ulcerative colitis (UC) and Crohn’s disease (CD)) predispose to cancer formation in the long run [46]. Inositol and its derivatives, as natural compounds, have shown a significant effect on inhibiting inflammation and carcinogenesis [47]. Here, we highlight the experimental results and mechanisms on inhibition of colitis and its induced carcinogenesis by *myo*-inositol and InsP_6_.

### 5.1. Effects of Inositol and InsP_6_ on Inhibiting Dextran Sulfate Sodium (DSS)-Induced Ulcerative Colitis (UC) and Carcinogenesis in Rodents 

DSS-induced colitis and carcinogenesis in mice is a useful experimental model for investigating the biology underlying colitis and colitis-associated carcinogenesis. Moreover, iron deficiency is commonly identified in DSS-induced chronic colitis. In addition to receiving low dose, cyclic, long-term DSS treatment for inducing chronic active colitis (to mimic flare-up and flare down activity of UC in human), mice also received a twofold dietary iron supplementation. Twofold dietary iron promoted colorectal carcinoma incidence from approximately 19% in mice with normal diet to approximately 88% in high-iron diet [46,47,48,49,50,51].

We have previously reported an inhibitory effect of inositol and InsP_6_ on colitis and colitis-induced carcinogenesis. In this study, we administered *myo*-inositol and InsP_6_ to DSS-induced colitis mice in drinking water and found a significant inhibition of colitis-induced cancer development as evaluated by reduction in tumor incidence, multiplicity, and volume [47]. Further studies have demonstrated that *myo*-inositol and InsP_6_ significantly suppressed chronic active colitis in non-tumor colorectal mucosa. A semiquantitative measurement using the pathological features of ulceration, reactive hyperplasic epithelial change, glandular distortion, and lymphoplasma cell infiltration, together called the active ulcerative colitis indices was performed. Mice treated with *myo*-inositol and InsP_6_ significantly inhibited the active ulcerative colitis indices, particularly inhibitied ulcer formation and the area of inflammation, in comparison to the colon of mice treated with DSS [47]. Immunohistochemical analyses further identified that treatment of mice with *myo*-inositol or InsP_6_ significantly reduced the number of Mac3+ macrophages in noninflamed and inflamed mucosa in mice. Further, *myo*-inositol and InsP_6_ treatment reduce inflammation-caused nitro-oxidative damage as measured using an immunohistochemical approach for detecting nitrotyrosine, a biomarker of nitro-oxidative stress [47,48,50,51,52]. Nitrotyrosine immunostaining showed strong reactivity in the macrophages in noninflamed and inflamed mucosa of the colon in DSS-induced UC mice. Nitrotyrosine staining intensity and nitrotyrosine-positive cell numbers in both noninflamed and inflamed areas of the colon were significantly decreased in the mice treated with *myo*-inositol [47,52]. These results strongly suggest that inositol’s observed prevention of colitis-induced carcinogenesis may occur in part secondary to reducing inflammation-induced nitro-oxidative stress.

Recent studies have shown that dietary *myo*-inositol significantly reduces stem/progenitor cell activation in colitis using phosphorylated beta-catenin^S552^ as a biomarker of recurrent dysplasia [22,53]. Importantly, immunohistochemical staining for beta-catenin^S552^ can be used to discriminate between ulcerative colitis patients with a history of low-grade dysplasia (LGD) from those without LGD. In a pilot clinical study, *myo*-inositol administration to two patients with a history of ulcerative colitis and low-grade dysplasia (LGD), resulted in a reduction in the number of activated intestinal stem cells observed compared to a placebo-treated control patient [53]. These data warrant large colitis-associated cancer chemoprevention trials, particularly, using colonic stem/progenitor biomarker beta-catenin^S552^.

### 5.2. Potential Mechanism of Inositol and Its Phosphates against Inflammation and Carcinogenesis

Inositol and its phosphates are the crucial intracellular signaling molecule/s [11,14,19,21]. In mammals, InsP_6_ plays diverse roles in the maintenance of homeostasis for storage of phosphate, while simultaneously functioning as an antioxidant and neurotransmitter [3,14,19,20]. InsP_6_ and other inositol phosphates including the pyrophosphates play critical role in cellular pathways involved in controlling cell proliferation and differentiation, RNA export, DNA repair, energy transduction, ATP regeneration, antioxidant, etc. [3,14,19,20]. By enabling communication between nucleus, cytoplasm, and external environment, InsP_6_ and the other inositol phosphates play a crucial role in many aspects of cell biology [3,14,19,20]. 

#### 5.2.1. PI3K Signaling and Gut Epithelial Progenitor Cell

Previous studies have demonstrated that loss of PI3K signaling markedly reduced the amount of gut epithelial progenitor cell expansion and dysplasia in an IL-10 deficient mouse model of colitis [22]. Animals lacking PI3K signaling in this model showed reduced downstream AKT and beta-catenin activation [22]. Importantly, a recent study suggests that *myo*-inositol treatment blocks activation of AKT and beta-catenin in a DSS-induced colitis model [53]. Specifically, mice receiving *myo*-inositol showed a 73% reduction in nuclear pAkt (11 ± 1-fold to 3 ± 1-fold, *p* = 0.006) and a 59% reduction in pβ-cat levels (45 ± 14-fold to 18 ± 4-fold *p* = 0.04) relative to DSS-treated controls [53]. Furthermore, in a small group of UC patients with known low-grade dysplasia, treatment with *myo*-inositol for 90 days reduced the amount of intestinal crypts containing more than 2 pβ-cat positive relative to patients receiving placebo (19% in the controls and 5% in *myo*-inositol treated) [53]. Combined, these results support the notion that *myo*-inositol treatment reduces colonic stem cell activation. However, more studies are required to determine if this effect is a cell autonomous effect on colonic stem cells or secondary to a reduction in local inflammation.

#### 5.2.2. Antioxidant

Inositol and its phosphates, specifically InsP_6_ have antioxidant properties, the latter functioning as a very potent antioxidant. This is thought to involve the chelation of iron and thus precluding its generation of free radicals through the Fenton reaction. This hypothesis is supported by experiments demonstrating that InsP_6_ inhibits •OH generation and subsequent lipid peroxidation driven by ascorbic acid and iron [54,55]. The precise mechanism how inositol itself acts as an antioxidant is not clear; however, it could be by phosphorylation to inositol phosphates. Indeed, Inositol hexaphosphate-citrate, a new molecule that retains the basic properties of both citric acid and InsP_6_ with increased valence states, demonstrates enhanced antioxidant activity [2]. Owing to their antioxidant properties, these molecules may have important biological functions beneficial to human and animal health, as well as industrial and environmental applications.

#### 5.2.3. Effects of Inositol on the Mucosal and Anti-Tumor Immunity

Inositol and its derivatives are known to have profound effects on a variety of immune cells. For example, InsP_6_ enhances inflammatory cytokine production in macrophages [56]. Similarly, neutrophils treated with InsP_6_ had a greatly augmented oxidative burst and generated higher levels of inflammatory cytokines in response to numerous bacterial stimuli [57,58]. Interestingly, InsP_6_ had opposing effects on inflammatory cytokine release in colonic epithelial cells [59,60,61]. These studies give rise to the intriguing possibility that InsP_6_ in the intestinal milieu may exert immunomodulatory effects on colonic epithelium under physiological conditions or during microbe-induced infection and/or inflammation in order to maintain the healthy colonic mucosa required to counteract infection.

In addition to maintaining intestinal barrier integrity and regulating innate immune cell function, studies have found that InsP_6_ can impact malignant cell growth through a cell autonomous modulation of immune cells. In mice and rats that had developed carcinogen-induced tumors, treatment with InsP_6_ results in significantly enhanced natural killer (NK) cell activity and tumor killing compared to untreated control animals [62,63]. These initial studies raise an interesting dichotomy for InsP_6_ in immune modulation. In response to bacterial insults inositol derivatives may induce strong immune responses thus resulting in successful clearance and prevention of a chronic inflammatory response, while maintaining barrier integrity during homeostatic conditions. However, in the setting of malignant transformation inositol may promote anticancer immune activity. Further studies are needed to elucidate the context-specific functions of inositol derivatives in the immune system in order to optimize anti-inflammatory and antitumor effects of inositol.

#### 5.2.4. Effects of Inositol and InsP_6_ on Cell Survival, Proliferation, and Differentiation

A basic alteration in cancer cells is the abnormal or uncontrolled cell proliferation. This abnormal growth in some cases seems to occur following tumor cell selection of a dedifferentiated or abnormally immature phenotype. Importantly, InsP_6_ and inositol can both reduce the tumor cell proliferation rate and induce differentiation. These observations have been made in several cancer cell types including colon, erythroleukemia, mammary, and rhabdomyosarcoma cells [40]. In colon cancer cells, InsP_6_ inhibits the expression of Galactose-N-acetyl-D-galactosamine (Gal-GalNAc), the most common cancer marker [64]. Importantly, InsP_6_ inhibits the expression of this marker in the intracellular mucus without suppressing the production of mucus, a normal function of colon epithelial cells.

#### 5.2.5. Effects of Inositol and InsP_6_ on DNA Damage Repair

InsP_6_ function as an antimutagenic agent has been recently demonstrated by Ra Yoon et al. [65]. As previously discussed, inositol and its derivatives have antioxidant activity, which, in part, may play a role in limiting DNA damage. However, inositol derivatives are also known to impact the function of the DNA damage repair machinery. For instance, InsP_6_ has been demonstrated to stimulate nonhomologous end-joining. Specifically, it has been proposed that this occurs due to binding of InsP_6_ to DNA end-binding protein Ku. This binding results in activation of DNA-PK a key modulator of nonhomologous end-joining repair [66,67,68,69]. Further studies have demonstrated that InsPs modulate the chromatin remodeling complexes SWI⁄SNF and INO80, which are required for successful DNA repair and transcriptional activation [70,71]. In addition, following high levels of DNA damage, cells activate cellular stress and prosurvival pathways including PI3K, AP-1, and NFkB, which are thought to result in malignant transformation. InsP_6_ has been shown to block epidermal growth factor-mediated activation of PI3K and AP-1 suggesting that along with enhancing DNA repair, inosine derivatives may prevent cellular adaption to environmental stresses that result in malignant transformation [72]. Taken together, these studies highlight the multifaceted role inositol derivatives play in DNA damage repair. 

### 5.3. Pharmacokinetics of InsP_6_

Owing to the charged nature of inositol hexaphosphoric acid and the lack of digestive enzyme phytase (that is required to hydrolyze the inositol-phosphate linkages), phosphorus and inositol in InsP_6_ are not considered general as bioavailable to humans or to nonruminant animals. However, the following factors have to be taken into account: (a) the naturally occurring state of InsP_6_ is the calcium–magnesium inositol hexaphosphate; (b) there are mechanisms within the cells such as pinocytosis to handle charged particles, even if the naturally occurring InsP_6_ was indeed highly charged; and (c) the presence of bacterial phytase in the organisms of the microbiome. Several studies in rodents and humans have now demonstrated that InsP_6_ is rapidly absorbed from the gastrointestinal tract, distributed through the plasma to various organs including the brain, and excreted from the lungs via exhaled air and through the kidneys via urine as inositol, InsP_1–5_ and InsP_6_. Interestingly, InsP_6_ crosses the blood-brain barrier and distributes to the brain [5,13,28,73,74,75,76]. It also is important to note that at least in mice, InsP_6_ efficiently crosses cell membranes and enter into the red blood cells.

It is also been demonstrated that neoplastic cells can rapidly take up InsP_6_ both in vitro and in vivo and precipitously dephosphorylate it to various lower inositol phosphates [3,5,64,77]. The relative proportion of these lower inositol phosphates varies greatly in different types of cells; thus, InsP_6_ may result in distinct alterations in different cell types. Overall, it is now clear that InsP_6_ is rapidly absorbed and distributed widely throughout the mammalian system. Additional research into the methodology to follow these fascinating molecules and their derivatives, as well as the application of these methodologies for research and practical application for human and animal health is needed [1,74,78,79,80,81].

## 6. Summary, Conclusions, and Future Direction

Inositol and its phosphates are natural dietary components that have known beneficial effects on human health. Particularly exciting is their potential role for cancer chemoprevention. From the nutritional perspective, monitoring inositol and its concentration in the blood will be important for the development of diets with high *myo*-inositol contents that could be used to evaluate and monitor its therapeutic and/or prophylactic efficacy in various diseases. Furthermore, longitudinal studies of circulating and cell type-specific inositol concentrations will also be important to determine the relationship between dietary *myo*-inositol intake and disease development such as in the case of cancer, diabetes mellitus, neuropsychiatric diseases, etc. This review highlights the strong experimental and observational evidence linking inositol and its phosphates to the prevention and treatment of colitis and colitis-induced carcinogenesis. Collectively, these studies demonstrate the profound biological role inositol and its phosphates play in diverse cellular processes and emphasize the importance of future research and clinical trials.

Since inflammatory bowel disease is a longstanding chronic active inflammatory bowel disease (IBD) with significantly increased risk for colorectal cancer and our previous study indicates that inositol displays a strong effect to inhibit IBD-induced carcinogenesis, we further propose the potential mechanism of inositol on inhibiting IBD-induced multiple stage carcinogenesis (from initiation to promotion, finally to invasive cancer), as seen in Figure 1. We proposed that IBD-caused free radicals, IP3-mediated AKT signaling, phospholipase A2/inositol-arachidonic acid metabolic pathway, and active eicosanoids are key biologic events involved in DNA damage/genetic instability, cell proliferation, and angiogenesis. Thereby, these events contribute to IBD-induced carcinogenesis, and as shown in Figure 1, inositol will prevent IBD-induced cancer development through blocking these events. As seen in Figure 1, exogenous inositol will participate in an intracellular inositol pool (a) to block IP3/AKT signaling and (b) to inhibit PGE2 and LTB4 production by blocking phospholipase A2/inositol-arachidonic acid release and COX2 and to inhibit inflammation-caused nitro-oxidative stress.

Looking toward the future, a potential nutritional deficiency of inositol and its phosphates in humans is a critical issue. It is essential to have critical data on the baseline level of inositol and its phosphates in humans and to understand the nutritional relationship between diet and inositol. It is also critical to understand the effect of nutritional deficiencies in inositol and its phosphates on human health, particularly, on cardiovascular disease, cancer, and chronic inflammatory disease (such as IBD). Although there are clear experimental data indicating the beneficial effect of inositol supplementation on chemoprevention of cancer and other disease, it is important to have a randomized and placebo-controlled clinical trial to further demonstrate the preventive effects for human diseases, particularly, for cancer.

## Figures and Tables

**Figure 1 molecules-26-00031-f001:**
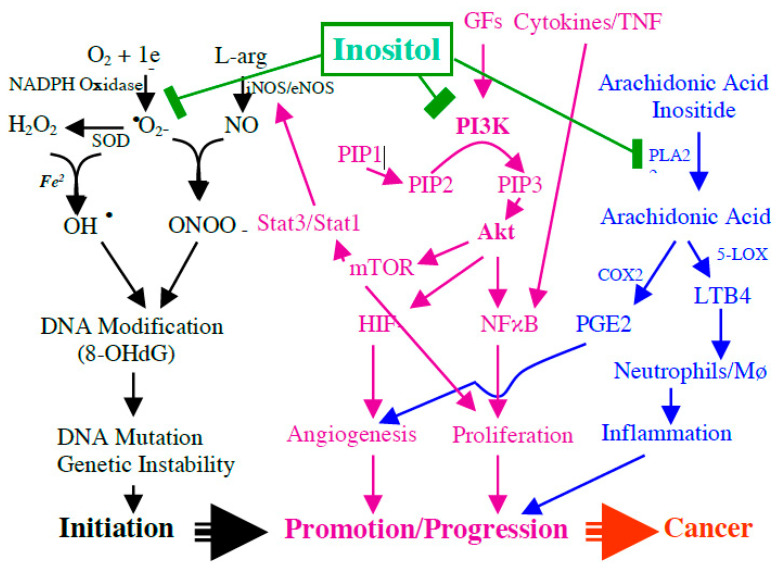
Potential mechanism of inositol on blocking the key biologic events in inflammatory process in inflammatory bowel disease (IBD), including IBD-caused free radicals, *myo*-inositol (1,4,5)-trisphosphate (IP3)-mediated AKT (protein kinase B) signaling, phospholipase A2/inositol-arachidonic acid metabolic pathway, and active eicosanoids, and blocking these key biologic events is crucial in inhibiting DNA damage/genetic instability, cell proliferation, and angiogenesis, and IBD-induced carcinogenesis.

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
