# Peer review of "Overview of Inositol and Inositol Phosphates on Chemoprevention of Colitis-Induced Carcinogenesis"

_molecules, 2020, doi:10.3390/molecules26010031_

Round 1

Reviewer 1 Report

The review by S. E. Weinberg et al. focuses on inositol and inositol phosphates derivatives, describing their role in different cellular outputs, including colitis, carcinogenesis and immune system function.

The review is interesting and portrays many aspects involving inositol and InsPs and their uptake by diet.

A question for the authors: why did the authors not include any image in the manuscript?

I would also recommend to add References in the following parts to have a better coverage of the literature:

1) Lanes 47-48 (add ref about PI and PIPs)

2) Lanes 73-72 (see above)

3) Lanes 85-86 (PLC and DAG/InsP3 production)

4) Lanes 111-116 (no ref here)

5) Lanes 221-222 (antioxidant role of InsPs)

Please also improve English, the text is full of mistakes and/or typos that must be corrected. Hereafter few examples:

TITLE (!!!!): Chemoprvention

Lane 51-52 'involving'

Lane 75-76 'Predominant'

Lane 86-87 'that involve'

Lane 107 'produces'

Lane 137 'Mechaniscally'

Lanes 138-139-139

Lane 235 'thru'

Lane 283 'demonstreated'

and so on...

Pending these recommendations, I would recommend the paper for publication on Molecules.

Author Response

Response to Reviewer 1:

why did the authors not include any image in the manuscript?

Reply: Thanks for reviewer’s suggestion. We have included a figure for potential mechanism of inositol on inhibiting IBD-induced carcinogenesis

I would also recommend to add References in the following parts to have a better coverage of the literature

Reply: These references have been added as reviewer’s suggested areas.

Please also improve English, the text is full of mistakes and/or typos that must be corrected.

Reply: We have thoroughly gone through and revised the manuscript with significant improvement in English.

Reviewer 2 Report

The study by Weinberg et al. proposed a role for inositol and inositol phosphates on chemoprevention of colitis-induced carcinogenesis.

The topic of the study is interesting. However, the review looks more like a list of information than a critical analysis of the literature: Authors just enunciate work made by others, instead of critically analysing it.

At the same time, conclusion section does not provide future research directions.

Moreover, the absence of figures makes it more difficult to understand the text. A final figure depicting the effects of inositol and inositol phosphates should be added.

Finally, there are many spelling errors in the text.

I suggest a major revision of the manuscript according to the previous comments.

Author Response

Response to Reviewer 2:

The topic of the study is interesting. However, the review looks more like a list of information than a critical analysis of the literature: Authors just enunciate work made by others, instead of critically analyzing it.

We have thoroughly gone through and revised the manuscript with significant improvement in analyzing the literature information.

At the same time, conclusion section does not provide future research directions.

Thanks for reviewer’s suggestion. The future research directions have been added.

The absence of figures makes it more difficult to understand the text. A final figure depicting the effects of inositol and inositol phosphates should be added.

Thanks for reviewer’s suggestion. We have included a figure for potential mechanism of inositol on inhibiting IBD-induced carcinogenesis

Finally, there are many spelling errors in the text.

We have thoroughly gone through and revised the manuscript with significant improvement in English.

Round 2

Reviewer 2 Report

The quality of the manuscript is much improved as a result of the revision, and the authors have dealt with my comments and suggestions to my satisfaction in this version. As far as I am concerned, the manuscript is now acceptable to be published.